# Effects of Telephone Aftercare Intervention for Obese Hispanic Children on Body Fat Percentage, Physical Fitness, and Blood Lipid Profiles

**DOI:** 10.3390/ijerph16245133

**Published:** 2019-12-16

**Authors:** Carlos Garza, David A. Martinez, Jihyung Yoon, Brett S. Nickerson, Kyung-Shin Park

**Affiliations:** 1College of Art and Science, Texas A & M International University, Laredo, TX 78041, USA; carlos.e.garza@colorado.edu; 2Rosenberg School of Optometry, University of the Incarnate Word, San Antonio, TX 78209, USA; David.Alonso.Mtz@gmail.com; 3College of Dentistry, New York University, NewYork, NY 10010, USA; dbswlgud0410@naver.com; 4College of Nursing and Health Sciences, Texas A & M International University, Laredo, TX 78041, USA; brett.nickerson@tamiu.edu

**Keywords:** aftercare intervention, childhood obesity, body composition, physical fitness, blood lipid profiles, telephone intervention

## Abstract

We investigated effects of 10-month telephone aftercare intervention following primary obesity intervention on changes in body fat percentage, physical fitness, and lipid profiles in obese Hispanic children. Seventy-one obese children were randomly assigned to (1) primary intervention and 10-month telephone aftercare intervention (PITI; *N* = 26), (2) primary intervention only (PI; *n* = 25), and (3) control (CON; *N* = 20). Anthropometric data, physical fitness, and blood samples were obtained before (PRE) and after (POST) eight-week primary intervention, and 10-month telephone aftercare intervention (1YEAR). Eight weeks of primary intervention significantly reduced body fat percentage, total cholesterol, triglycerides, and low-density lipoprotein (LDL-C) with increases in VO_2max_, flexibility, muscular strength, and HDL-C (PITI and PI, *p* < 0.05). 1YEAR measurements returned to baseline for the PI whereas those measurements in PITI remained significantly different when compared to PRE (*p* < 0.05). CON observed negative changes in all variables at POST, which were improved slightly during the subsequent school year. Levels of cholesterol, triglycerides, and LDL-C are correlated to changes in body fat percentage, suggesting that fat loss is effective in preventing and managing obesity-related disorders. Results indicate that telephone intervention is an effective aftercare in stabilizing positive changes obtained from a short-term intensive intervention.

## 1. Introduction

Research has shown that obesity rates have increased dramatically in adults and children over the last 30 years. Moreover, obesity is recognized as a major cause of morbidity and mortality [1,2]. Although poor health outcomes associated with obesity arise in the later stages of life, their causes can be developed during childhood [1,3]. This is especially true for minority children in the United States of America [2,3,4]. For instance, the trends for obesity have shown a high prevalence in Hispanic groups in the United States, which is even more profound in children who reside in low-income households. Consequently, the alarmingly high incidence rate of obesity puts this population at high risk for developing future health problems [4].

Summer break represents approximately 23% of the typical American school year and is the largest consecutive period for out-of-school time [5]. Unfortunately, positive changes in body composition and physical fitness achieved during the school year have shown to stall or revert during the summer months due to physical inactivity and ease of access to food [6,7]. For these reasons, recent studies have revealed that summer camps can be an effective strategy for helping reduce body fat percentage and improve physical fitness [8,9,10,11].

Serum lipid abnormalities are associated with obesity [12] and known to be positively correlated with higher risks of coronary heart disease, cardiovascular disease, and hypertension [12,13,14,15]. Studies that have implemented a dietary and physical activity intervention have reported significant reductions in low-density lipoprotein (LDL-C), triglycerides (TG), and total cholesterol (TC) [16,17]. However, research has yet to report changes in lipid profiles in children over the summer break.

Many studies have reported short-term success in the treatment of obesity; however, long-term success is not commonly evaluated as follow-up measurements are seldom carried out [18]. Only one known study reported a reversion to baseline values in body weight and body mass index 10 months after a summer camp. The reversion back to baseline was attributed to the difficulty with adhering to dietary and physical activity programs [18]. A few studies have utilized a telephone aftercare intervention following a primary obesity intervention in order to prevent weight regain, which commonly occurs. Nonetheless, results regarding body composition, physical fitness, and blood lipids have been inconclusive [19,20,21]. Therefore, further research is required to achieve a better understanding of the effects of the telephone aftercare intervention following a primary obesity intervention.

Through investigations of previous research, we presumed that subjects who completed an eight-week primary intervention (summer camp) would show significant improvements in body composition, physical fitness, and blood lipid profile compared to the control. Furthermore, we also hypothesized that those who were provided a 10-month telephone aftercare intervention would be able to stabilize their post-camp results compared to those who did not receive telephone calls. To investigate these hypotheses, the present study evaluated the effectiveness of a 10-month telephone intervention as an aftercare intervention to a primary eight-week summer camp intervention on body composition, physical fitness, and blood lipid profiles.

## 2. Materials and Methods

### 2.1. Subjects

A total of 87 subjects aged 10–14 years old (4–8th grade), from the underprivileged Hispanic community in southern Texas, participated in this study. Subjects were recruited through local newspaper advertisement for a university summer camp program in two consecutive summers in years 2012 and 2013.

Seventy-one subjects completed the entire collection process. Subjects were considered eligible if their body mass index (BMI) was over the 85th percentile as determined by the centers for disease control and prevention (CDC) BMI percentile calculator for child and teen [22]. Eligible subjects for each gender were randomized into one of three groups: (1) Eight-week primary intervention with 10-month telephone aftercare intervention (PITI, *n* = 33), (2) eight-week summer camp only (PI, *n* = 32), and (3) control (CON, *n* = 22). For a fair randomization, compensation was paid to the subjects in the control group instead of providing interventions as written in the consent form.

A total of 16 subjects were excluded due to missing more than 5 days of camp or missing one of the data collection days. Therefore, 26 PITI (14 boys and 12 girls), 25 PI (13 boys and 12 girls), and 20 CON (11 boys and 9 girls) completed this study. Compensation was paid to the subjects in the control group instead of providing interventions as written in the consent form. The experimental protocol (TAMIU IRB 2011–04–05) was approved by the Institutional Review Board, and subjects and their parents provided written informed assent and consent, respectively.

### 2.2. Primary Intervention: 8 Weeks of Summer Camp

The primary obesity intervention was held five days a week between 1:00 and 5:00 p.m. for 8 consecutive weeks during summer break. Subjects participated in four 50-min training sessions every day: Aerobic exercise (70–80% HR max), muscular strength and endurance, Zumba dance with stretching (50–80% HR max), and fun activities (such as tag games, modified ball games, dodge ball, basketball, etc.). Subjects were asked to exercise at low to moderate intensity during aerobic exercise and Zumba dance. For that, all subjects were required to wear a heart rate monitor (Polar, Oy, Finland) and were required to rest if heart rate increased above the target heart rate zone. An hour of nutrition education and an hour of behavioral counseling sessions were given to subjects once every week. A certified dietitian calculated the total daily energy expenditure for all subjects using the equation for overweight boys and girls aged 3–18 years [23] and provided a weekly diet plan during the 8 weeks of summer camp. A pedometer (Omron, PA, USA) was provided to each individual to calculate weekly steps.

The parents of PITI and PI groups attended two parental education meetings before the start of summer camp. These meetings provided parental education tools regarding avoidance of emotional eating and setting behavior goals of nutrition and exercise, especially concerning healthier food choices. Furthermore, parents were enlightened on the importance of promoting self-confidence and positive self-body image.

### 2.3. Ten-month Telephone Aftercare Intervention

After 8 weeks of summer camp, a 10-month telephone aftercare intervention was given to subjects and their parents in the PITI group; however, no intervention was given to the PI group during follow-up. A pedometer (Omron, PA, USA) was provided to each subject in PITI to calculate weekly steps during the 10-month telephone aftercare intervention. The telephone aftercare was provided once a week, and each phone call took approximately 10–20 min. Subjects assigned to PITI were asked if they reached a goal of 10,000 steps/day and kept the diet pattern from the summer camp. It was not required to follow guidelines of physical activities and diet pattern, but they were encouraged to continue physical activities and diet from summer camp. After that, parents were asked to report weekly steps recorded in the pedometer and communicate with investigators to verify if subjects maintained the recommended calorie intake and physical activity. All phone calls were conducted by a single trained investigator. No weekly diet plans were provided during this aftercare intervention.

### 2.4. Measurements

Subjects were required to visit the laboratory three times: Prior to attending the summer camp (PRE), immediately after camp (POST), and 10 months (1YEAR) after the end of camp. For each day of visit, subjects arrived at the testing center between 8:00 and 09:00 a.m. after a minimum of 10 h of fasting and underwent blood sampling, anthropometric measurements, and physical fitness tests.

Subjects’ anthropometric measurements were obtained twice by a trained technician, with measurements of height and weight being calculated to the nearest 0.1 cm and 0.1 kg, respectively. During height and weight measurements, subjects wore indoor clothes and no shoes.

#### 2.4.1. Body Composition and Physical Fitness

Body fat percentage was determined using bioelectrical impedance analysis (BIA: Quantum IV, RJL systems, MI, USA). Subjects were required to have at least 10 h fasting and void their bladder before the BIA test to minimize measurement errors. During BIA measurement, subjects were asked to lie supine with arms 30° from the body and legs not touching. Electrodes were placed on the dorsal surface of the right hand and foot, following the manufacturer’s guidelines. Resistance and reactance were measured to determine body fat percentage using the company-provided software.

Subjects also had physical fitness assessed. Cardiovascular function (VO_2max_) was estimated via the Rockport one-mile walk test [24] in which subjects walked on the treadmill at the fastest rate possible. Flexibility was determined via the sit-n-reach as described in American College of Sports Medicine (ACSM) guidelines [25]. An adjustable mechanical hand dynamometer (Lafayette Instrument, Lafayette, IN, USA) was used to assess hand grip strength. The test was conducted with the subject’s dominant hand in standing position.

#### 2.4.2. Blood Collection and Measurement

Blood collection: 5 mL of venous blood was drawn at three separate times throughout the study: Before the summer camp, 48 h after the last exercise session of summer camp, and after the 10 months of telephone aftercare. Blood samples were centrifuged in plain tubes at 1000 × g for 15 min (Allegra X-15R Refrigerated Centrifuge, Beckman Coulter, Irving, TX, USA) to get serum sample.

Blood lipid profiles: Commercial kits were used for all biochemical determinations and manufacturers’ recommendations were followed throughout. An enzymatic assay (Genesys 10 Bio, Thermo Fisher Scientific, Waltham, MA, USA) with commercial kits (Pointe Scientific Inc. Lincoln Park, MI, USA) was used to perform standard enzymatic measurements of triglyceride concentration, total cholesterol, and high-density lipoprotein cholesterol (HDL-C) at 500 nm. Low-density lipoprotein cholesterol (LDL-C) were calculated using Friedewald et al. formula [26].

### 2.5. Statistical Analyses

Sample size was estimated using the operating characteristic curve [27]. Power calculations were conducted to assess the overall changes in body composition and total cholesterol within the group using results from Farris et al. [28], which indicated that 20 subjects in each group were needed to detect significant changes in body composition and total cholesterol through an intervention program with 80% power at α = 0.05. All statistical analyses were conducted using Sigmaplot 13 (Systat Software, inc., San Jose, CA, USA). Two-way analysis of variance (ANOVA) with repeated measures was used to analyze changes in body composition, physical fitness, and blood lipid profiles during the summer camp as well as the 10-month telephone aftercare intervention. Post hoc tests using a Tukey post hoc test were performed when appropriate. Significance for all comparisons was set at an alpha level of <0.05.

## 3. Results

The gender differences were not analyzed in the present study because the number of boys and girls in each group (PITI: 14 boys and 12 girls; PI: 13 boys and 12 girls; CON: 11 boys and 9 girls) were smaller than the required sample size based on the power analysis (i.e., *n* = 20 at 80% power). However, combining boys and girls together did not cause any group differences because we stratified boys and girls evenly among groups. We conducted two-way repeated measures ANOVA with a Tukey post hoc test because variables in this study passed the Shapiro–Wilk normality test and Brown–Forsythe equal variance test. Results of these two tests indicate that there is no baseline difference found in all variables measured in this study. Significant time and interaction effects were found in all variables except age.

Changes in physical characteristics and physical fitness at baseline (PRE) after eight-week summer camp (POST) and after 10-month aftercare (1YEAR) are shown in Table 1. There were no significant differences found among the three groups at PRE. Height at 1YEAR significantly increased as compared to PRE in all three groups (*p* < 0.01). Changes in body weight showed a different pattern in all three groups. In CON group, body weight significantly increased at POST (*p* = 0.027) and 1YEAR (*p* = 0.006) as compared to PRE. Body weight in PITI decreased at POST (*p* = 0.024). An increase in body weight was observed at 1YEAR compared to POST (*p* = 0.014); however, body weight at 1YEAR was not significantly greater than PRE (*p* = 0.327). PI showed a significant decrease in body weight at POST (*p* = 0.031); however, it significantly increased at 1YEAR as compared to POST (*p* = 0.002), which was greater than PRE (*p* = 0.020). CON observed significant differences at POST as compared to PITI (*p* = 0.011) and PI (*p* = 0.09) at 1YEAR as compared to PITI (*p* = 0.026). BMI decreased at POST in both treatment groups (PITI: *p* = 0.004, PI: *p* = 0.007), while CON observed an increase at POST (*p* = 0.031). From POST to 1YEAR, the PI saw an increase in BMI (*p* = 0.023), but PITI did not show any change in body fat percentage, indicating PITI successfully maintained the reduction in body fat percentage from eight weeks of summer camp for another 10 months (PRE vs. 1YEAR: *p* = 0.016). BMI in CON was significantly greater than PITI (*p =* 0.016) and PI (*p* < 0.021) at POST and PITI (*p* = 0.044) at 1YEAR.

Eight weeks of summer camp significantly decreased body fat percentage and increased estimated VO_2max_, flexibility, and muscular strength in both PITI and PI. The 1YEAR measurements returned to PRE value for PI, whereas the values for PITI remained statistically significant for up to 10 months with the telephone aftercare intervention.

When compared to PRE, a significant decrease in body fat percentage was observed at POST in both treatment groups (PITI: *p* = 0.005, PI: *p* = 0.007), while body fat percentage in CON significantly increased at POST (*p* = 0.028). From POST to 1YEAR, the PI saw a significant increase in body fat percentage (POST vs. 1YEAR: *p* = 0.017). PITI did not see a significant change in body fat percentage between POST and 1YEAR; therefore, PITI successfully maintained the improvement from eight weeks of summer camp for another 10 months (PRE vs. 1YEAR: *p* = 0.021). Along with this, a significant difference was observed in body fat percentage between PITI and PI at 1YEAR (*p* = 0.037) and CON vs. PITI (*p* = 0.005) and PI (*p* = 0.009) at POST and CON vs. PITI (*p* = 0.021) at 1YEAR.

Similar to the changes in body fat percentage, both PITI and PI saw a significant increase in estimated VO_2max_ between PRE and POST (PITI: *p* = 0.008, PI: *p* = 0.013). However, PI saw a significant reduction in VO_2max_ from POST to 1YEAR (POST vs. 1YEAR: *p* = 0.011), while no significant change was found in PITI, which thus reinforces the idea that telephone aftercare intervention is a successful secondary intervention. VO_2max_ in CON significantly decreased at POST (PRE vs. POST: *p* = 0.041) and returned to PRE value through the school year (PRE vs. 1YEAR: *p* = 0.89). CON observed significant differences at POST as compared to PITI (*p* = 0.023) and PI (*p* = 0.019) as well as at 1YEAR as compared to PITI (*p* = 0.034).

Both PITI and PI groups showed significant improvement in flexibility between PRE and POST (PITI: *p* = 0.019, PI: *p* = 0.024). PITI maintained this improvement from the summer camp over 10 months (PRE vs. 1YEAR: *p* = 0.028); however, PI observed recess to baseline (PRE) in the 10 months after the summer camp. A significant difference in flexibility at 1YEAR was found between PITI and PI (*p* = 0.037). In CON, flexibility significantly decreased during summer camp (PRE vs. POST: *p* = 0.041). Flexibility was greater in two intervention groups at POST (PITI: *p* = 0.037; PI: *p* = 0.025) and greater in PITI at 1YEAR (*p* = 0.046) as compared to CON.

Muscular strength measured using the handgrip dynamometer significantly increased at POST (PITI: *p* = 0.044, PI: *p* = 0.025) and 1YEAR (PITI: *p* = 0.009, PI: *p* = 0.007) as compared to PRE. Muscular strength of CON was unaltered at POST (*p* = 0.275); however, it increased at 1YEAR (*p* = 0.036). CON showed a significantly lower value at 1YEAR as compared to PITI (*p* = 0.041) and PI (*p* = 0.031).

Changes in blood lipid profiles are displayed in Table 2. Eight weeks of summer camp significantly changed blood lipid profiles in all three groups. Both summer camp intervention groups (PITI and PI) showed significant decreases in levels of total cholesterol, triglycerides, and LDL-C and an increase in HDL-C, whereas the CON group showed adverse results except for HDL-C. After 10-month telephone aftercare intervention (1YEAR), PITI maintained these improvements of blood lipid profiles from the summer camp, while PI saw a significant reversion in blood lipid profiles, almost back to PRE value.

## 4. Discussion

The purpose of the present study was to investigate whether a 10-month telephone aftercare intervention could stabilize changes in body fat percentage, physical fitness, and blood lipid profiles obtained after a primary obesity intervention in obese Hispanic children. We observed that eight weeks of primary care (i.e., summer camp) helped induce positive changes in physical fitness and blood lipid profiles in the two intervention groups (PITI and PI). Moreover, the 10-month telephone aftercare intervention helped subjects in PITI successfully maintain their improvements during this follow-up intervention period. In contrast, measurements reverted back to pre-summer camp values for PI who did not receive the 10-month telephone aftercare.

Previous studies noted that physical inactivity and ease of access to food during summer break often resulted in children undergoing a decline in physical fitness and an increase in weight gain [29], notably in Hispanic and African American children [30]. Our findings are in agreement with the aforementioned studies. Specifically, the control group in the present study underwent negative changes in all parameters of body composition, physical activity, and blood lipid profiles over summer break.

Results of the current study show that subjects who completed the eight-week summer camp observed reductions in body weight, BMI, and body fat percentage. These results are similar with previous studies that implemented the use of a lifestyle intervention program over summer break and found significant reductions in body fat percentage [8,10,16]. For example, Huelsing et al. [8] explored the use of a summer camp for obese children where group educational sessions and structured physical activities were provided to campers. The campers ultimately reported statistically significant reductions in body weight, BMI, and systolic blood pressure. Thus, these findings further indicate the benefits of summer lifestyle intervention camps.

Engaging in physical activity is essential because an increased level of physical activity can help reduce obesity-related complications, even when weight loss is not present [20]. Evidence of improvements in physical fitness following a summer lifestyle intervention has been observed in recent studies in which subjects experienced improvements in one-mile run times [8], curl-ups [9], aerobic fitness [10], cardiorespiratory fitness, and overall physical fitness [11,31]. Findings of the present study support earlier work, which reported that subjects who completed a summer camp experienced significant improvement in their physical fitness characteristics (e.g., estimated maximal oxygen uptake, flexibility, and handgrip strength).

Summer campers in the present study observed positive changes in blood lipid variables. This is similar to results of previous lifestyle intervention treatments, which helped significantly reduce levels of total cholesterol [28], LDL-C, and triglycerides [16,17]. Our results also correspond with a meta-analysis conducted on seventeen childhood obesity prevention program studies, which concluded that diet and/or physical activity programs had beneficial effects on HDL-C and LDL-C [32]. It was also reported that a significant reduction in triglycerides were associated with a decrease in adiposity [32], which further supports results of the present study where changes in body fat percentage were found in association with total cholesterol, LDL-C, and triglycerides.

Although summer camps and other short-term obesity studies have produced positive results, long term follow-up measurements are seldom carried out. One previous study reported that summer campers experienced reversions to initial BMI, body weight, and body fat percentage 10 months after the summer camp [18]. Results from the PI group in the present study are similar to these aforementioned findings. In contrast, PITI successfully maintained their post-camp improvements in body composition and physical fitness until 10 months after the conclusion of the summer camp.

Telephone interventions have exhibited beneficial results such as significant weight loss and increased physical activity when used as the primary intervention [33,34,35]. However, effects of using a telephone intervention as an aftercare are still inconclusive. Although benefits were observed in the telephone aftercare intervention group (i.e., PITI) in the present study, previous research has reported conflicting results. Specifically, two studies that administered a long-term telephone aftercare intervention observed no significant differences in body composition [19] and physical activity [20] between intervention and control groups. For example, Wing et al. [19] utilized a telephone aftercare following a six-month obesity intervention and concluded that changes in body composition were likely not induced by telephone aftercare as both control and intervention groups noticed similar weight loss. Ströbl et al. [20] also implemented a telephone aftercare to a primary obesity intervention and reached a similar conclusion (i.e., intervention and control groups noticed similar positive changes in physical activity).

A study carried out by Hyman et al. [21] found that a six-month telephone aftercare intervention was beneficial in maintaining changes in cholesterol. Schiel et al. [36] also implemented 12 months of telephone/email intervention following a hospitalized weight reduction program for obese children and reported that subjects observed weight loss from the primary intervention and weight stabilization during the telephone aftercare intervention.

Obese children are at risk of developing harmful, short- and long-term health effects such as dyslipidemia, a primary indicator of metabolic syndrome [17,35,37,38]. Therefore, the maintenance of cholesterol and triglyceride levels in children is vital in reducing the prevalence of obesity-related disorders. To the authors’ knowledge, this is the first research experiment to measure changes in blood lipids after a long-term telephone aftercare preceded by a summer camp. At the conclusion of our telephone intervention, those who received telephone aftercare intervention exhibited maintenance in their total cholesterol, triglycerides, HDL, and LDL cholesterol. The non-telephone intervention group did not achieve the same positive results as they saw a significant reversion to pre-camp values, thus strengthening the suggestion that the use of telephone intervention in our study was effective in preventing long-term regression and further demonstrating the advantages of telephone intervention as a secondary intervention.

There are two potential limitations to this study that must be considered. Firstly, because the aftercare intervention was done through telephone, parents and subjects were simply asked whether they kept up with the recommended diet and pedometer steps. Therefore, there was difficulty in strictly controlling diet and exercise as the subjects’ report might not be accurate and truthful. Secondly, because our study was conducted in a predominantly Hispanic area and focused solely on Hispanic children, it cannot be fully concluded whether telephone intervention as a follow-up to a primary intervention is effective in maintaining positive results in other demographics. As a result, future research that administers the use of a telephone aftercare should do so with strict diet and exercise control to ensure the effectiveness of telephone follow-up. Additionally, future telephone aftercare studies should be conducted in areas with diverse demographics to assess differences in results between varying groups of individuals.

## 5. Conclusions

Implementing a secondary intervention is important for continuing obesity-related improvements over a long period of time as changes observed following short-term interventions are shown to revert shortly after an intervention is over. Our findings suggest that a telephone intervention is a desirable aftercare technique that can be implemented in future obesity interventions, especially when significant improvements during primary interventions are seen. This is an important factor that health and fitness professionals should be aware of when working with obese populations. Telephone aftercare intervention is especially beneficial for school children as its low cost and ability to reinforce intervention strategies allow subjects to maintain their dietary adherence and physical activity throughout the school year. Additionally, the aspect of parental involvement provides children with structure and support as well as motivation to succeed. Furthermore, because primary interventions are too intensive to conduct during the school year, implementing a telephone aftercare provides school children a nonexhaustive approach to maintaining positive health changes seen after primary intervention.

## Figures and Tables

**Table 1 ijerph-16-05133-t001:** Physical characteristics and physical fitness of subjects at baseline (PRE), after eight-week primary intervention (POST), and after 10-month follow-up (1YEAR).

Variables	Control (*n* = 20)			PITI (*n* = 26)			PI (*n* = 25)		
	PRE	POST	1YEAR	PRE	POST	1YEAR	PRE	POST	1YEAR
Age (year)	12.1 ± 0.33			12.0 ± 0.34			11.9 ± 0.36		
Height (cm)	151.9 ± 2.6	152.1 ± 2.7	155.9 ± 3.5 **	152.5 ± 2.5	152.7 ± 2.7	156.7 ± 3.9 **	151.8 ± 2.5	151.9 ± 2.6	156.0 ± 4.7 **
Weight (kg)	62.7 ± 2.3	64.6 ± 2.5 *	67.2 ± 3.3 **#	62.9 ± 2.5	60.6 ± 2.8 *+	64.2 ± 3.6 #+	62.3 ± 2.5	60.4 ± 2.9 *+	65.9 ± 3.6 *##
BMI	27.8 ± 0.6	28.7 ± 0.7 *	28.4 ± 0.7	27.6 ± 0.5	26.5 ± 0.6 **+	26.8 ± 0.7 *&+	27.5 ± 0.5	26.7 ± 0.6 **+	27.7 ± 0.7 #
Body fat percentage (%)	31.5 ± 1.0	32.8±1.1 *	32.4 ± 1.1	31.3 ± 1.1	27.9 ± 1.3 **++	29.1 ± 1.3 *&+	31.0 ± 1.0	28.1 ± 1.4 **++	30.5 ± 1.2 #
VO2max (ml/kg/min)	41.3 ± 1.7	39.5 ± 1.9 *	41.6 ± 2.0	40.9 ± 1.6	45.7 ± 1.7 **++	45.2 ± 1.8 **+	41.2 ± 1.7	45.9±1.8*++	43.4 ± 2.0#
Flexibility (cm)	28.4 ± 1.4	26.1 ± 1.7 *	27.8 ± 1.8	27.6 ± 1.5	31.9 ± 1.6*+	31.3 ± 1.8 *&+	28.9 ± 1.5	32.7 ± 1.7 *+	29.3 ± 1.8
Grip strength (kg)	23.7 ± 1.8	23.1 ± 2.3	25.7 ± 2.1	24.5 ± 1.7	26.7 ± 1.8 *+	27.9 ± 1.9	22.7 ± 1.8	25.7 ± 1.9 *+	27.4 ± 2.0

Note: Values are means ± SE. PITI, primary intervention + 10-month telephone intervention; PI, primary intervention only; *, significantly different from PRE (*p* < 0.05); **, significantly different from PRE (*p* < 0.01); #, significantly different from POST (*p* < 0.05); ##, significantly different from POST (*p* < 0.01); &, different from PI at 1YEAR; +, significantly different from CON at the same time point (*p* < 0.05); ++ significantly different from CON at the same time point (*p* < 0.01).

**Table 2 ijerph-16-05133-t002:** Lipid profiles at baseline (PRE), after eight-week primary intervention (POST), and after 10-month follow-up (1YEAR).

Variables	Control (*n* = 20)			PITI (*n* = 26)			PI (*n* = 25)		
Unit:mg∙dL^−1^	PRE	POST	1YEAR	PRE	POST	1YEAR	PRE	POST	1YEAR
Total cholesterol	147.4 ± 4.1	156.3 ± 4.5	151.3 ± 4.1	146.2 ± 3.8	135.7 ± 3.9	139.2 ± 3.7	143.9 ± 3.6	136.5 ± 3.7	140.4 ± 3.6
		*p* = 0.019 *	*p* = 0.034 #		*p* = 0.015 *	*p* = 0.036 *		*p* = 0.018 *	*p* = 0.035 #
					*p* = 0.019 +	*p* = 0.037 +		*p* = 0.023 +	
Triglycerides	124.7 ± 4.4	130.7 ± 4.4	127.4 ± 4.0	126.7 ± 4.1	96.8 ± 4.7	106.9 ± 4.5	122.3 ± 4.5	98.7 ± 4.7	116.6 ± 4.6
		*p* = 0.026 *			*p* = 0.004 **	*p* = 0.024 *		*p* = 0.006 **	*p* = 0.014 #
					*p* = 0.008 ++	*p* = 0.031 +		*p* = 0.014 +	
HDL-C	33.4 ± 1.6	33.2 ± 1.9	33.8 ± 1.6	32.7 ± 1.3	44.9 ± 1.7	40.2 ± 1.4	34.6 ± 1.5	42.1 ± 1.3	37.9 ± 1.4
					*p* = 0.005 **	*p* = 0.036 *		*p* = 0.008 **	*p* = 0.030 #
					*p* = 0.010 +	*p* = 0.042 +		*p* = 0.013 +	
LDL-C	89.1 ± 3.5	97.0 ± 4.0	94.6 ± 3.8	88.2 ± 3.5	71.4 ± 3.3	77.6 ± 3.7	84.8 ± 3.5	74.7 ± 3.8	84.6 ± 3.6
		*p* = 0.019 *	*p* = 0.043 *		*p* = 0.003 **	*p* = 0.029 *		*p* = 0.009 **	*p* = 0.011 #
					*p* = 0.007 ++	*p* = 0.041 # *p* = 0.033 +		*p* = 0.019 +	

Note: Values are means ± SE. PITI, primary intervention + 10-month telephone intervention; PI, primary intervention only; HDL-C, high-density lipoprotein cholesterol; LDL-C, low-density lipoprotein cholesterol; *, significantly different from PRE (*p* < 0.05); **, significantly different from PRE (P < 0.01); #, significantly different from POST (*p* < 0.05); +, significantly different from CON at the same time point (*p* < 0.05); ++ significantly different from CON at the same time point (*p* < 0.01).

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
