# Peer review of "Effects of Telephone Aftercare Intervention for Obese Hispanic Children on Body Fat Percentage, Physical Fitness, and Blood Lipid Profiles"

_ijerph, 2019, doi:10.3390/ijerph16245133_

Round 1

Reviewer 1 Report

In this article the effect of telephone aftercare was investigated in maintaining weight loss in obese adolescents.

Overweight and obesity have several medical and psychosocial implications. Adverse implications of weight problems during childhood track into adulthood. Furthermore, the financial cost of obese children that grow up to obese adults is immense for society. Tackling weight problems in childhood is therefore important for both children’s and adult’s health and for society. However, existing weight loss treatments have only limited success, specifically at long-term. The scientific community has an important responsibility in research on maintaining post-treatment weight, which will have impact on the physical and psychological well-being and will reduce the long-term risks of morbidity and mortality. A telephone aftercare program seems a promising tool to maintain post-treatment weight.

 The study seems correctly performed, yet the manuscript can be further inproved. This is supported by the following remarks.

General remark

Please ask the advice of a native speaker. The readability of the text can be improved.

Introduction

“”Through investigation of previous research, we hypothesized that an 8-week primary intervention (summer camp) would significantly reduce % body fat along with improvements in physical fitness and blood lipid profile as compared to the control. Furthermore, we also hypothesized that those who were provided a 10-month telephone aftercare intervention would be able to stabilize their post-camp results compared to those who did not receive telephone calls” power analysis should be mentioned in the methods section in detail.

Materials and methods

When was the study performed 2017-2018? Or earlier. Please mention this timeframe.

Children who had a BMI greater than the 85th percentile were included; these children have overweight. Where there also children included with a BMI >95th percentile? The title of the manuscript should be adapted to “overweight Hispanic children” or “overweight and obese Hispanic children” accordingly.

Telephone aftercare intervention: could this be described in more detail? This makes the study unique. How long did the phone calls last? How many people did the phone calls? Was it always the same investigator who called the same patient?

Please summarize the paragraph on physical fitness significantly. Not all formulas should be mentioned completely, you can refer to a paper describing the formula e.g.

Blood collection and measurement: This paragraph can also be shortened significantly. Not all technical information should be mentioned; e.g. the type of -80C freezer is not relevant

Results

This paragraph should be completely rewritten. In the present form all the results from the tables are just repeated.

You could start with possible baseline differences in the 3 groups. How many boys and girls participated in the different groups? Are these groups comparable?

Why isn’t BMI mentioned in the tables? Certainly in growing children BMI is more important than weight alone.

It seems that the only parameter that remained significantly better after 1 year in the PITI group compared to the other groups and compared to baseline is %body fat! This seems disappointing. Weight increases also in the PITI group after 1 year, so what is the clinical relevance? Is there a statistical effect on BMI???

Can the results from physical fitness, flexibility and strength be mentioned in a table?

Discussion

The discussion usually starts with summarizing the results from the present study. The following information should be mentioned in the introduction: “23% of the ordinary American school year is devoted to summer break, which is the longest time spent consecutively out of class for school children [5]. Physical inactivity and ease of access to food during summer often results in children undergoing a decline in physical fitness and an increase in weight gain [29]. It was reported in a systemic review that children observed an accelerated weight gain during the summer break which was not seen during the school year, notably in Hispanic and African American children [30]. This is consistent with a previous study in which children who underwent a nine months’ lifestyle physical education class witnessed that their improvements in body composition, cardiovascular fitness, and fasting insulin levels were reversed back during the summer break, further illustrating summer as a time of physical inactivity and overconsumption of food.

After calculating the effects of intervention on BMI, the clinical relevance should be addressed in the limitations section!

Author Response

Thank you so much for giving me a chance to revise my manuscript. I believe I did my best to follow suggestions in review. But, due to limitation in this study or limitation in my understanding, it may not satisfy all requests or raise other questions. Please let me know if you have any questions in my revision. Below you will find my answers (red color) to each of your comments or questions. Below you will find my responses to your review in blue color.

In this article the effect of telephone aftercare was investigated in maintaining weight loss in obese adolescents.

Overweight and obesity have several medical and psychosocial implications. Adverse implications of weight problems during childhood track into adulthood. Furthermore, the financial cost of obese children that grow up to obese adults is immense for society. Tackling weight problems in childhood is therefore important for both children’s and adult’s health and for society. However, existing weight loss treatments have only limited success, specifically at long-term. The scientific community has an important responsibility in research on maintaining post-treatment weight, which will have impact on the physical and psychological well-being and will reduce the long-term risks of morbidity and mortality. A telephone aftercare program seems a promising tool to maintain post-treatment weight.

 The study seems correctly performed, yet the manuscript can be further inproved. This is supported by the following remarks.

 General remark

Please ask the advice of a native speaker. The readability of the text can be improved.

One of co-author, Dr. Nickerson who is a native speaker reviewed this manuscript and made changes as advised.

 Introduction

“”Through investigation of previous research, we hypothesized that an 8-week primary intervention (summer camp) would significantly reduce % body fat along with improvements in physical fitness and blood lipid profile as compared to the control. Furthermore, we also hypothesized that those who were provided a 10-month telephone aftercare intervention would be able to stabilize their post-camp results compared to those who did not receive telephone calls” power analysis should be mentioned in the methods section in detail.

It is described under 2.5. Statistical analyses as shown below;

“Sample size was estimated using the operating characteristic curve [27]. Power calculations were conducted to assess the overall changes in body composition and total cholesterol within the group using results from Farris et al. [28] which indicated that 20 subjects in each group were needed to detect a significant changes in body composition and total cholesterol through an intervention program with 80% power at α = 0.05.”

Materials and methods

When was the study performed 2017-2018? Or earlier. Please mention this timeframe.

This study was conducted before 2017 as I added under 2.1. Subjects. We finished data analyses in year 2015, but the publication has been delayed since the first author moved into other University for his further study.

Children who had a BMI greater than the 85th percentile were included; these children have overweight. Where there also children included with a BMI >95th percentile? The title of the manuscript should be adapted to “overweight Hispanic children” or “overweight and obese Hispanic children” accordingly.

That is because we advertised that overweighed or obese children would be recruited in the local newspaper. However, we found that only three subjects’ (out of 71 subjects) BMI were under 95 percentile from the screening tests, but these three subjects’ BMI were also close to 95 percentile. So, we decided to describe these subjects “obese” in this manuscript. Since not all subjects’ BMI were over 95 percentile, we described the eligibility as we advertised in the newspaper. If needed, I can put “overweight” in the title and other sentences.

Telephone aftercare intervention: could this be described in more detail? This makes the study unique. How long did the phone calls last? How many people did the phone calls? Was it always the same investigator who called the same patient?

I added changes as requested. Since the purpose of telephone calls was just to motivate subjects and their parents (we did not try to control their lifestyle neither to collect any additional data from the telephone calls), questions were very simple as described in the paragraph.

Please summarize the paragraph on physical fitness significantly. Not all formulas should be mentioned completely, you can refer to a paper describing the formula e.g.

It was summarized as requested. It looks better after removing equations.

Blood collection and measurement: This paragraph can also be shortened significantly. Not all technical information should be mentioned; e.g. the type of -80C freezer is not relevant

Thank you so much for pointing it out. It was shortened as requested and it is in better shape.

Results

This paragraph should be completely rewritten. In the present form all the results from the tables are just repeated.

You could start with possible baseline differences in the 3 groups. How many boys and girls participated in the different groups? Are these groups comparable?

There was no baseline differences found in three groups. This is added at the beginning of Results. The number of boys and girls were described in the first paragraph in Results also. Since eligible subjects for each gender were stratified evenly to each group, we believe these groups were comparable. Since number of boys and girls in each group was smaller than the sample size we found from the power analysis, we did not compare between boys and girls in this study.

Why isn’t BMI mentioned in the tables? Certainly in growing children BMI is more important than weight alone.

This is added to second paragraph under 3 Results and also added in the table 1. and in the second paragraph in Results.

It seems that the only parameter that remained significantly better after 1 year in the PITI group compared to the other groups and compared to baseline is %body fat! This seems disappointing. Weight increases also in the PITI group after 1 year, so what is the clinical relevance? Is there a statistical effect on BMI???

I apologize first that I did not indicate other differences (for example between two intervention groups vs. control) in this manuscript. I had difficulty to put all these group differences in the tables and also I thought most important thing in this study was to see if the telephone aftercare can prevent weight regain. Therefore, I only marked group differences between two intervention groups (PITI and PI) at each measurement time (one received telephone aftercare and the other group not). But, after getting your review, I understood why all other differences should be indicated in the manuscript. I added these differences in paragraphs and tables to avoid confusion. Thank you so much for pointing out.

Can the results from physical fitness, flexibility and strength be mentioned in a table?

The figure 1 was removed from the manuscript and data for physical fitness was added to table 1. So, there are two tables and no figure in this manuscript.

Discussion

The discussion usually starts with summarizing the results from the present study. The following information should be mentioned in the introduction: “23% of the ordinary American school year is devoted to summer break, which is the longest time spent consecutively out of class for school children [5]. Physical inactivity and ease of access to food during summer often results in children undergoing a decline in physical fitness and an increase in weight gain [29]. It was reported in a systemic review that children observed an accelerated weight gain during the summer break which was not seen during the school year, notably in Hispanic and African American children [30]. This is consistent with a previous study in which children who underwent a nine months’ lifestyle physical education class witnessed that their improvements in body composition, cardiovascular fitness, and fasting insulin levels were reversed back during the summer break, further illustrating summer as a time of physical inactivity and overconsumption of food.

This part in Discussion was changed as recommended. The purpose and results of this study were mentioned at the beginning of Discussion. Also, the rest part mentioned above were removed from discussion (23%.......).

After calculating the effects of intervention on BMI, the clinical relevance should be addressed in the limitations section!

As mentioned above, the differences between CON vs. two intervention groups in each measurement point were added. This is not caused by the limitations of this study, but just because authors thought marking all of these were not necessary to emphasize effect of telephone intervention following the primary intervention on prevention of weight regain (yo-yo syndrome). I apologize it again.

Reviewer 2 Report

Comments

This is a very interesting manuscript. I enjoyed reading it. However, I have some comments that I think can help the author(s) to improve it.

Overall:

The author(s) need to edit the manuscript extensively for grammatical and language errors. I encountered a couple of these in the entire manuscript

Beginning sentences with abbreviations / acronyms / numbers may need to be changed. It is best to write the terms/words in full at the beginning of the sentences to improve the readability of the manuscript.

In the introduction:

Perhaps author(s) need to explain whether they are writing about minority children “in the US”, and also be specific about the region where the research was conducted. Leaving the sentence on line 37 of the manuscript as “This is especially true for minority children” may be confusing to someone who is from another continent / or an African country, where “minority groups” are mainly Caucasians. Moreover, if the country/area of research is not described, it becomes difficult to make proper assumptions about the outcomes of the specific research. Describing the area of research also make it easy for the reader to generalize the outcomes to other settings with similar characteristics.

In the methods section:

Author(s) also need to also describe the population they researched. This will help readers in making proper assumptions about the outcomes of this research. This will also improve the understandibility of their outcomes. It will also be useful if the author(s) show the actual calculation of the sample size. I acknowledge that they gave a brief statement on this in the statistical analysis. My other understanding is that sample size calculation needs to be presented under the “study subject / population section” not under the “statistical analysis section”. It looks like this was convenient sampling. However, describing the population is needed and indicating who was the researchers’’ target audience – aside that author(s) indicated that they sought to target Hispanic children - the age / school grades etc. of the students also need to be presented in this section. A short statements on the description of the control population (CON) will be useful after the descriptions the author(s) gave for the 8 week and 10mo interventions.

In the results section:

In Table 1, author(s) started including the age of the children. As much as this is useful, the other sociodemographic characteristics of the participants should be presented both in the methods and results sections. These variables are important since they contribute to degree of outcomes that can be observed during participating in an intervention. The abbreviations SUTI and SU used in Figure 1 are confusing. Perhaps they should be changed to “PITI and PI” to correspond with these terms used in the figure legends. Moreover, PITI and PI are suitable because they are the main abbreviations used in the Tables and right through the manuscript narrative.

Author Response

Thank you so much for giving me a chance to revise my manuscript. I believe I did my best to follow suggestions in review. But, due to limitation in this study or limitation in my understanding, it may not satisfy all requests or raise other questions. Please let me know if you have any questions in my revision. Below you will find my answers (red color) to each of your comments or questions. Below you will find my responses to your review in blue color.

Comments

This is a very interesting manuscript. I enjoyed reading it. However, I have some comments that I think can help the author(s) to improve it.

Overall:

The author(s) need to edit the manuscript extensively for grammatical and language errors. I encountered a couple of these in the entire manuscript

I asked one of co-authors (Dr. Nickerson who is a native speaker) to check this manuscript and we did our best to correct them.

Beginning sentences with abbreviations / acronyms / numbers may need to be changed. It is best to write the terms/words in full at the beginning of the sentences to improve the readability of the manuscript.

Full descriptions of abbreviations were added. For example, body mass index (BMI), centers for disease control and prevention (CDS).

In the introduction:

Perhaps author(s) need to explain whether they are writing about minority children “in the US”, and also be specific about the region where the research was conducted. Leaving the sentence on line 37 of the manuscript as “This is especially true for minority children” may be confusing to someone who is from another continent / or an African country, where “minority groups” are mainly Caucasians. Moreover, if the country/area of research is not described, it becomes difficult to make proper assumptions about the outcomes of the specific research. Describing the area of research also make it easy for the reader to generalize the outcomes to other settings with similar characteristics.

Thank you so much for pointing out. I totally forgot the description is only applied for USA population. I added “United States of America” in that sentence to clarify the region where the research was conducted.

In the methods section:

Author(s) also need to also describe the population they researched. This will help readers in making proper assumptions about the outcomes of this research. This will also improve the understandibility of their outcomes. It will also be useful if the author(s) show the actual calculation of the sample size. I acknowledge that they gave a brief statement on this in the statistical analysis. My other understanding is that sample size calculation needs to be presented under the “study subject / population section” not under the “statistical analysis section”. It looks like this was convenient sampling. However, describing the population is needed and indicating who was the researchers’’ target audience – aside that author(s) indicated that they sought to target Hispanic children - the age / school grades etc. of the students also need to be presented in this section. A short statements on the description of the control population (CON) will be useful after the descriptions the author(s) gave for the 8 week and 10mo interventions.

Changes added under 2.1. Subjects section. A total of 87 subjects aged 10-14 year old (4-8th grade), from the underprivileged Hispanic community in southern Texas, participated in this study. Subjects were recruited through local newspaper advertisement as one of university summer camp programs in two consecutive summers in years 2012 and 2013.

In the results section:

In Table 1, author(s) started including the age of the children. As much as this is useful, the other sociodemographic characteristics of the participants should be presented both in the methods and results sections. These variables are important since they contribute to degree of outcomes that can be observed during participating in an intervention. The abbreviations SUTI and SU used in Figure 1 are confusing. Perhaps they should be changed to “PITI and PI” to correspond with these terms used in the figure legends. Moreover, PITI and PI are suitable because they are the main abbreviations used in the Tables and right through the manuscript narrative.

I realized that the importance of sociodemographic data correlated with the outcomes measured in this study after finishing data analyses. Unfortunately, the sociodemographic characteristics of the participants were not collected in this study. I collected data regarding socio-economic status in more recent studies, but I could not do that for the present study.

I am really sorry for the confusion I made in the abbreviations. Actually I removed this figure and put them to table 1 which was requested by other reviewer. Anyway, they is no more confusion in new submission.

Reviewer 3 Report

The authors present an interesting study about the effects of telephone aftercare intervention on physical fitness and body composition in obese hispanic children. Its main strength is that the intervention to improve physical activity and body composition was undertaken in an interesting population, which is currently understudied and its consequences might be of public health interest.

The intervention resulted in improved several physical fitness and body composition parameters and the effect sizes were expectedly not large. There are, however, some aspects that need to be revised. Below are some comments in an attempt to improve the quality of the manuscript.

Introduction.

Sentence "This is especially true for minority children. For instance, the
prevalence of obesity in Hispanic population is greater as compared to non-Hispanic white." needs reference.

Methods.

Line 76. Although is quite obvious that CDC refers to Centers for Disease Control and Prevention, explanation for abbreviation is needed.

Regarding sit and reach test, this reviewers suggest the use of a different flexibility test, as one of the disadvantages of the sit and reach test is that not everyone is of the same stature. Some people have longer arms or legs than others and this can skew results when measuring the flexibility of the lower back and hamstrings.

For further research, this reviewer suggest the use of the non-dominant hand grip strength as a measure as well, as it could provide valuable information and a better muscular strength measure.

Discussion.

Line 298. Capital letter needed "they".

Author Response

Thank you so much for giving me a chance to revise my manuscript. I believe I did my best to follow suggestions in review. But, due to limitation in this study or limitation in my understanding, it may not satisfy all requests or raise other questions. Please let me know if you have any questions in my revision. Below you will find my answers (red color) to each of your comments or questions. Below you will find my responses to your review in blue color.

The authors present an interesting study about the effects of telephone aftercare intervention on physical fitness and body composition in obese hispanic children. Its main strength is that the intervention to improve physical activity and body composition was undertaken in an interesting population, which is currently understudied and its consequences might be of public health interest.

The intervention resulted in improved several physical fitness and body composition parameters and the effect sizes were expectedly not large. There are, however, some aspects that need to be revised. Below are some comments in an attempt to improve the quality of the manuscript.

Introduction.

Sentence "This is especially true for minority children. For instance, the
prevalence of obesity in Hispanic population is greater as compared to non-Hispanic white." needs reference.

This sentence was changed and also reference was added to the manuscript.

Methods.

Line 76. Although is quite obvious that CDC refers to Centers for Disease Control and Prevention, explanation for abbreviation is needed.

Full name of CDC added and the abbreviation was put in parenthesis.

Regarding sit and reach test, this reviewers suggest the use of a different flexibility test, as one of the disadvantages of the sit and reach test is that not everyone is of the same stature. Some people have longer arms or legs than others and this can skew results when measuring the flexibility of the lower back and hamstrings.

Thank you so much for the information. I recognized that my knowledge for measurement tools is very limited. I will definitely study tools for flexibility and apply it in my future studies.

For further research, this reviewer suggest the use of the non-dominant hand grip strength as a measure as well, as it could provide valuable information and a better muscular strength measure.

I will conduct a literature review for the measurement of muscle strength and will apply it to my next study. Since I appreciate again to pointing it out.

Discussion.

Line 298. Capital letter needed "they".

I am really sorry that I could not find it. I believe it was fixed by auto-correction.